# Tunable Phase Structure in Mn-Doped Lead-Free BaTiO₃ Crystalline/Amorphous Energy Storage Thin Films

**Jianlu Geng** [1], **Dongxu Li** [1], **Hua Hao** [1,2,*], **Qinghu Guo** [2], **Huihuang Xu** [1], **Minghe Cao** [1], **Zhonghua Yao** [1] **and Hanxing Liu** [1]

1 State Key Laboratory of Advanced Technology for Materials Synthesis and Processing, School of Material Science and Engineering, International School of Material Science and Engineering, Wuhan University of Technology, Wuhan 430070, China
2 Foshan Xianhu Laboratory of the Advanced Energy Science and Technology Guangdong Laboratory, Xianhu Hydrogen Valley, Foshan 528200, China
* Correspondence: haohua@whut.edu.cn

**Abstract:** For dielectric energy storage materials, high polarization and high breakdown strengths are a long-standing challenge. A modulating crystalline/amorphous phase structure strategy is proposed by Mn-doping and annealing temperature to enhance the energy storage performance of pure BaTiO₃ (BT) films. In this study, lead-free Mn-doped BT films were prepared on Pt/Ti/SiO₂/Si substrates via the sol-gel method, and the effects of the crystalline/amorphous phase ratio on polarization and electric properties were analyzed. A small amount of Mn-doping in BT could reduce the annealing temperature and enhance polarization with an Mn content of 8%. In addition, the energy storage properties of BT-8%Mn films achieve the best energy storage performance in terms of energy density and efficiency of 72.4 J/cm³ and 88.5% by changing the annealing temperature to 640 °C. BT-8%Mn energy storage films also possess good stability over a wide temperature range of 20 °C–200 °C, which demonstrates that crystalline/amorphous engineering is a simple and effective way to enhance energy storage applications of dielectric films.

**Keywords:** energy storage; crystalline/amorphous; thin film; annealing temperatures

## 1. Introduction

Dielectric capacitors, as a passive component, are widely used in advanced electric systems for energy storage and conversion, benefiting from high power densities, fast charging/discharging speeds, and excellent temperature/cycling capabilities [1–3]. Compared to other chemical power sources such as batteries and SOFCs (solid oxide fuel cells), inorganic dielectric materials, especially thin films, have received more attention because they exhibit high polarization characteristics [4,5]. However, the constricted relationship between polarization and breakdown strength also hinders the enhancement of the energy density of dielectric films. Therefore, there is a need to find a new way to break the current dilemma, which is to achieve high energy density while also meeting the demands of miniaturization and integration [6–8].

Barium titanate (BT) is a typical ferroelectric perovskite material and possesses the advantages of being lead-free, environmentally friendly, and having a high dielectric constant (300~350) at room temperature [9]. However, the low withstand voltage (1000 kV/cm) and high dielectric losses of pure BT films make it difficult to be applied [10,11].

In order to solve these problems, many strategies, such as interface engineering [12–14], chemical doping [15,16], domain structure regulation [17,18], crystalline/amorphous phase [19,20], etc., have been widely attempted. Among them, the crystalline/amorphous phase is utilizing the different advantages of high polarization in the crystalline phase and high breakdown strength in the amorphous phase to attain better energy storage properties [21,22]. For example, Yuyao Zhao et al. [23] prepared BT thin films on (100)-oriented

conductive perovskite $LaNiO_3$ substrates by magnetron sputtering at a low processing temperature of 350 °C. Due to the existence of the $LaNiO_3$ template layer, the (001)-oriented sputtering growth of thin films was promoted, and perovskite BT thin films composed of ultrafine columnar nanocrystals (the diameter of these nanocrystals can be controlled at about 10 nm) were successfully prepared. These ultrafine columnar nanocrystals with a small plane diameter and high filling density will cause highly limited and severely attenuated electric dipoles. This leads to stable and small remnant polarization and saturated polarization at high applied field strength, resulting in ultra-high energy storage density ($135 \pm 10$ J/cm$^3$) and efficiency ($80\% \pm 4\%$) and good thermal stability and cycle stability ($2 \times 10^8$ charge-discharge cycles) at 150 °C~170 °C. In addition, the work of adding excess-Ti source $Bi(Mg_{0.5}Ti_{0.5})O_3$-based films can achieve a high energy density of 126 J/cm$^3$ utilizing the crystalline temperature difference of materials [24]. In addition, Xuewen Jiang et al. [25] prepared 3% Mn-doped 0.94 BT-0.06 Bi ($Zn_{0.5}Zr_{0.5}$) $O_3$ thin films by the sol-gel method and also achieved a good energy density of 85 J/cm$^3$ at a breakdown strength of 4700 kV/cm by constructing hexagonal with high polarization and cubic polymorphic domain structures at a low annealing temperature of 600 °C, where rapid annealing is used (three layers of one annealing, a total of six layers). Mn ions can change between +2, +3, and +4 oxidation states [26], making it possible to reduce the oxygen vacancies in the BT film when it is doped as the acceptor element [27] ($Mn''_{Ti} + V_O^{\cdot\cdot} + \frac{1}{2}O_2 \rightarrow Mn_{Ti} + O_O$) [28], and also to inhibit the partial reduction in $Ti^{4+}$ to $Ti^{3+}$, reducing oxygen vacancies to enhance breakdown strength [29].

However, further research on hexagonal phase adjustments and their effect on polarization is insufficient. Combined with the research results of Gyung Hyun Ryu et al. [30], amorphous BT was prepared by pulsed laser deposition (PLD); in situ high-temperature X-ray diffraction patterns showed that the hexagonal BT phase appeared earlier than the cubic BT phase, and the electrical properties were improved by changing the annealing process. Huihuang Xu et al. [20] prepared $BaZr_{0.25}Ti_{0.75}O_3$ thin films by the sol-gel method. The microstructure of the coexistence of the crystalline phase and the amorphous phase was constructed by adjusting the annealing temperature. The goal of the crystalline phase providing the polarization value and the amorphous phase providing the withstand voltage value was achieved, and the high energy storage density of 60.8 J/cm$^3$ was also achieved. This shows that adjusting the annealing temperature is a direct method to adjust the phase structure in the film. Therefore, the effectiveness of Mn-doping and annealing temperature to adjust phase structure and polarization behavior on BT-based energy films needs to be further explored.

In this work, the strategy of constructing a heterogeneous structure with the coexistence of crystalline and amorphous phases in BT-based films by adjusting the annealing temperature (560 °C, 600 °C, 640 °C, 680 °C, 720 °C) is proposed to improve the energy storage density and efficiency. Meanwhile, the role of Mn-doping on oxygen vacancy and crystallinity is investigated to lower the annealing temperature of BT films. Finally, Mn-doped BT films were prepared on $Pt/Ti/SiO_2/Si$ substrates via the sol-gel method, and the phase composition, microstructure, energy storage, and dielectric properties were investigated. Energy density of 72.4 J/cm$^3$ and efficiency of 88.5% are achieved at an annealing temperature of 640 °C with 8% Mn-doping content.

## 2. Materials and Methods

In this study, Mn-doped BT films (BT-x%Mn films) were prepared on $Pt/Ti/SiO_2/Si$ substrates using the sol-gel method. First, a precursor solution with a concentration of 0.2 mol/L was configured; barium acetate ($Ba(CH_3COO)_2$, 99.0%, Sinopharm Chemical Reagent) and manganese acetate tetrahydrate ($Mn(CH_3COO)_2 \cdot 4H_2O$, 99.0%, Sinopharm Chemical Reagent) were weighed using an electronic balance according to their stoichiometric ratio and dissolved in acetic acid ($CH_3COOH$, 99.0%, Sinopharm Chemical Reagent) and recorded as liquid A. Tetrabutyl titanate ($Ti[OCH(CH_3)_2]_4$, 99.0%, Sinopharm Chemical Reagent) was dissolved in 2-methoxyethanol ($C_3H_8O_2$, 99.0%, Sinopharm Chemical

Reagent), noted as liquid B. Liquid A and liquid B were mixed and then stirred for 12 h to form a clear precursor solution. The precursor solution was filtered through a syringe filter to obtain a pure solution and left to age for 24 h to obtain a stable solution.

The precursor solution was spin-coated on the substrate at 5000 rpm for 30 s. After each coating, the film was heated at 200 °C to remove water and organic solvents, then at 350 °C and 450 °C, respectively, to make the films denser on the heating table in air, and finally annealed by rapid thermal processing to obtain an Mn-doped BT thin film in an air atmosphere. The platinum top electrode was deposited on the thin film with a diameter of 0.2 mm by magnetron sputtering to measure its electrical properties. In addition, pure BT films were prepared by the same preparation process and compared with doped films. Due to the best energy storage performance of 8%Mn-BT film at 640 °C, the annealing temperature of pure BT film is 640 °C.

The surface morphology of the Mn-doped BT films was measured using a scanning electron microscope (SEM, Zeiss Ultra Plus, Oberkochen, Germany), and the relevant parameters of the SEM equipment are as follows: secondary electron resolution: 1.0 nm (15 kV), 1.2nm (1 kV); the spectrum resolution is better than 127 eV at Mn $K_\alpha$. Electron microscope magnification: 12~1,000,000×; analysis element range: 4 Be–94 Pu. The surface roughness of the film and its changes are represented by the results of an atomic force microscope (AFM, NanoscopeIV, VEECO, Lateral resolution: <0.2 nm, Santa Barbara, CA, USA). The phase structure of Mn-doped BT films was studied by grazing incidence X-ray diffraction (GI-XRD, PANalytical, Almelo, The Netherlands) with a diffraction angle of 20~60°. The relevant parameters of the XRD equipment were: high-voltage generator power of 4 kW; maximum high-voltage of 60 kV; the maximum anode current of 60 mA; Cu target $K_\alpha$ radiation ($\lambda$ = 1.54056 Å) was used. The dielectric constant and losses of Mn-doped BT films were measured using an impedance analyzer (Agilent 4294, Santa Clara, CA, USA). The polarization electric-field loops (P-E) of Mn-doped BT films were measured at 1 kHz using a ferroelectric measuring system (CPE 1801, poly K, Philipsburg, PA, USA). The elemental composition and elemental chemical states of the films were tested using X-ray photoelectron spectroscopy (XPS, ESCALAB 250Xi, American Thermo Fisher, Waltham, MA, USA), and the parameters related to the XPS equipment are as follows: The energy scanning range is 0~5000 eV; the energy range is 1~400 eV; the beam spot range is 20~900 μm; analysis of element range: all elements except He; the energy resolution is 0.45 eV; sensitivity is 1.0 Mcps @ 0.6 eV (650 μm beam spot); the imaging spatial resolution is 3 μm [31].

For nonlinear dielectrics with certain energy dissipation, such as ferroelectrics, relaxor ferroelectrics, and antiferroelectrics, the charging energy storage density ($W_{st}$), and discharge energy storage density ($W_{rec}$) can be obtained by integrating the effective area between the polarization axis and the polarization-electric field (P-E) charge-discharge hysteresis loop curve. The charging energy storage density ($W_{st}$) and discharging energy storage density ($W_{rec}$) of dielectric capacitors can be calculated by the following equations:

$$W_{st} = \int_0^{P_{max}} EdP \tag{1}$$

$$W_{rec} = \int_{P_r}^{P_{max}} EdP \tag{2}$$

In the formula, $W_{st}$ is the charge energy storage density, $W_{rec}$ is the discharge energy storage density, $E$ is the applied electric field strength, $P$ is the polarization strength, $P_{max}$ is the maximum polarization strength, and $P_r$ is the residual polarization. In addition, the use of high-speed switching circuits through the pulse discharge current (dynamic method) has also proved to be another feasible method to measure the discharge energy storage density of the medium.

In energy storage applications, the charge-discharge efficiency ($\eta$) of the device cannot be ignored for dielectric capacitors. In general, in the process of discharge, the electrical

energy stored in the capacitor will be lost. The value of the energy loss density, represented by $W_{loss}$, is equal to the area of the *P-E* loop. Typically, the energy storage efficiency of a capacitor can be calculated using the following equations:

$$\eta = \frac{W_{rec}}{W_{st}} = \frac{W_{rec}}{W_{rec} + W_{loss}} \tag{3}$$

In the formula, $W_{st}$ is the charge energy storage density, $W_{rec}$ is the discharge energy storage density, $W_{loss}$ is the lost energy density, and $\eta$ is the energy storage efficiency.

## 3. Results and Discussion

Figure 1a,b shows the GIXRD diagrams of pure BT film and BT-8% Mn film at different annealing temperatures. As the annealing temperature increases, the diffraction peaks of pure BT films appear until 720 °C. However, for the BT-8%Mn films, the diffraction peaks of the perovskite phase appear at 680 °C, which demonstrates that Mn-doping could reduce the annealing temperature to some degree. Figure 1c shows the Mn 2p XPS pattern and its fitted curve in the BT-8%Mn film; the fitted peak at 641.3 eV corresponds to $Mn^{2+}$. With the increase in binding energy, the fitting peaks of $Mn^{3+}$ and $Mn^{4+}$ appear, which indicates that there are multivalent Mn ions in the films [32]. Figure 1d shows the changes in the O 1s XPS before and after Mn-doping. The main peak around 530 eV is usually considered to be lattice oxygen in the lattice. The additional shoulder peak around 531 eV could be considered an oxygen vacancy after excluding other possibilities (such as carbonate [33] or $Al_2O_3$ et al.) [34–36]. As there are no other miscellaneous oxides present in the thin film sample and the reduction in the intensity of the shoulder peak in these O 1s after Mn-doping, this shoulder peak represents the oxygen vacancy existence. It can be found that the introduction of Mn ions reduces the oxygen vacancies in the film.

Figure 1e,f shows the breakdown strength, polarization, and energy storage properties with different contents of Mn-doping, respectively. There is a tendency for the breakdown strength and polarization to increase with the increase in Mn-doping content, and the energy storage density also shows the same trend. The best performance of the films with an energy storage density of 54.12 $J/cm^3$ was achieved at 8% Mn-doping content, so 8% Mn-doping was chosen for the subsequent study of modulating the crystalline and amorphous phases by annealing temperature.

Figure 2 shows the SEM images of BT-8%Mn films annealed at different temperatures. It can be observed that the film surface is smooth and flat with only a very small number of grains at low annealing temperatures (560 °C, 600 °C). As the annealing temperature increases to 640 °C, an increase in the number and size of grains can be clearly observed. When the annealing temperature reaches 720 °C, there are almost crystallized grains in the film. This is because higher temperatures are beneficial for crystalline nucleation. As the annealing temperature increases, the film shows a reduction in the amorphous phase and an increase in the crystalline phase; this is consistent with the results in the GIXRD diagrams. Figure 2f shows the fracture of the BT-8% Mn film annealed at 640 °C with a thickness of approximately 200 nm.

Figure 3a–e shows the AFM images of the BT-8%Mn films annealed at different temperatures. The change in color depth in the graph represents the fluctuation of the surface [37]. At annealing temperatures of 560 °C, 600 °C, and 640 °C, bright spots can be seen scattered on the surface of the films and tend to increase with increasing temperature. These bright spots represent the grains in the film, but the film is still dominated by amorphous phases, which is consistent with the absence of diffraction peaks observed in GIXRD. On the contrary, when the annealing temperature is increased to 680 °C or 720 °C, the uneven surface of the film is covered with interconnected grains, indicating that the film is dominated by crystalline phases. The roughness of the film can reflect its crystallization and is often expressed in terms of root mean square roughness (RMS). The RMS of the films can be calculated by using nanoscope analysis software, as shown in Figure 3f. The RMS at 560 °C, 600 °C, 640 °C, 680 °C, and 720 °C are 0.4 nm, 0.621 nm, 0.889 nm, 1.91 nm,

and 2.34 nm, respectively. The trend in the RMS data show that roughness increases with annealing temperature. At 560 °C, 600 °C, and 640 °C, a low RMS indicates low roughness, flat surfaces, and amorphous dominance. As the annealing temperature increases to 680 °C and 720 °C, the high RMS demonstrates the dominance of the crystalline phase. The crystalline phase distribution situation in SEM is consistent with the change in roughness of AFM, which demonstrates the change in the crystallization of the film.

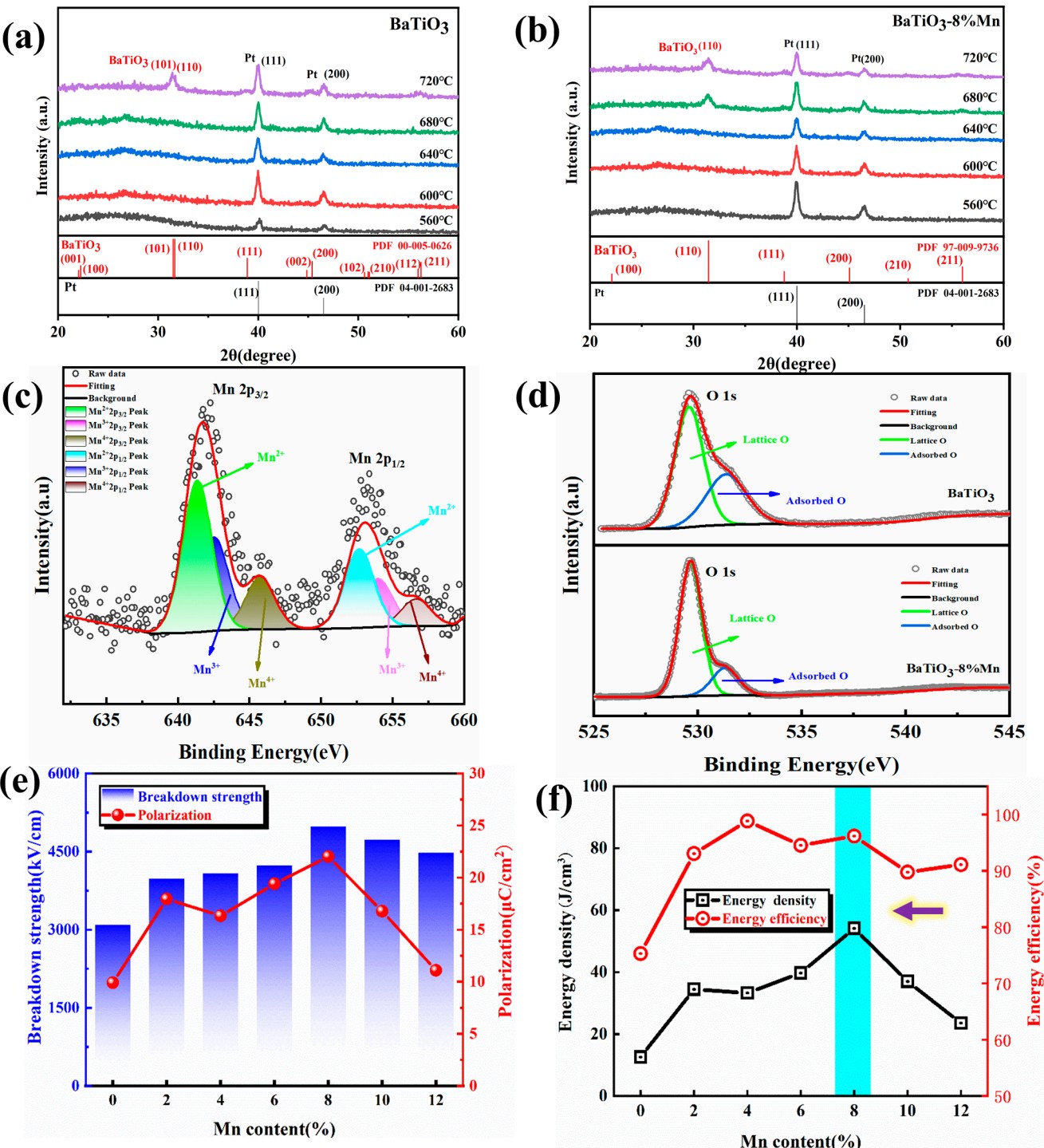

**Figure 1.** GIXRD spectra at different annealing temperatures: (**a**) BT films; (**b**) BT-8%Mn films; (**c**) Mn 2p XPS spectrum in BT-8% Mn films; (**d**) XPS spectrum of O 1s in BT and BT-8%Mn films; (**e**) polarization and breakdown strength of BT-x %Mn films; (**f**) energy storage performance of BT-x %Mn films.

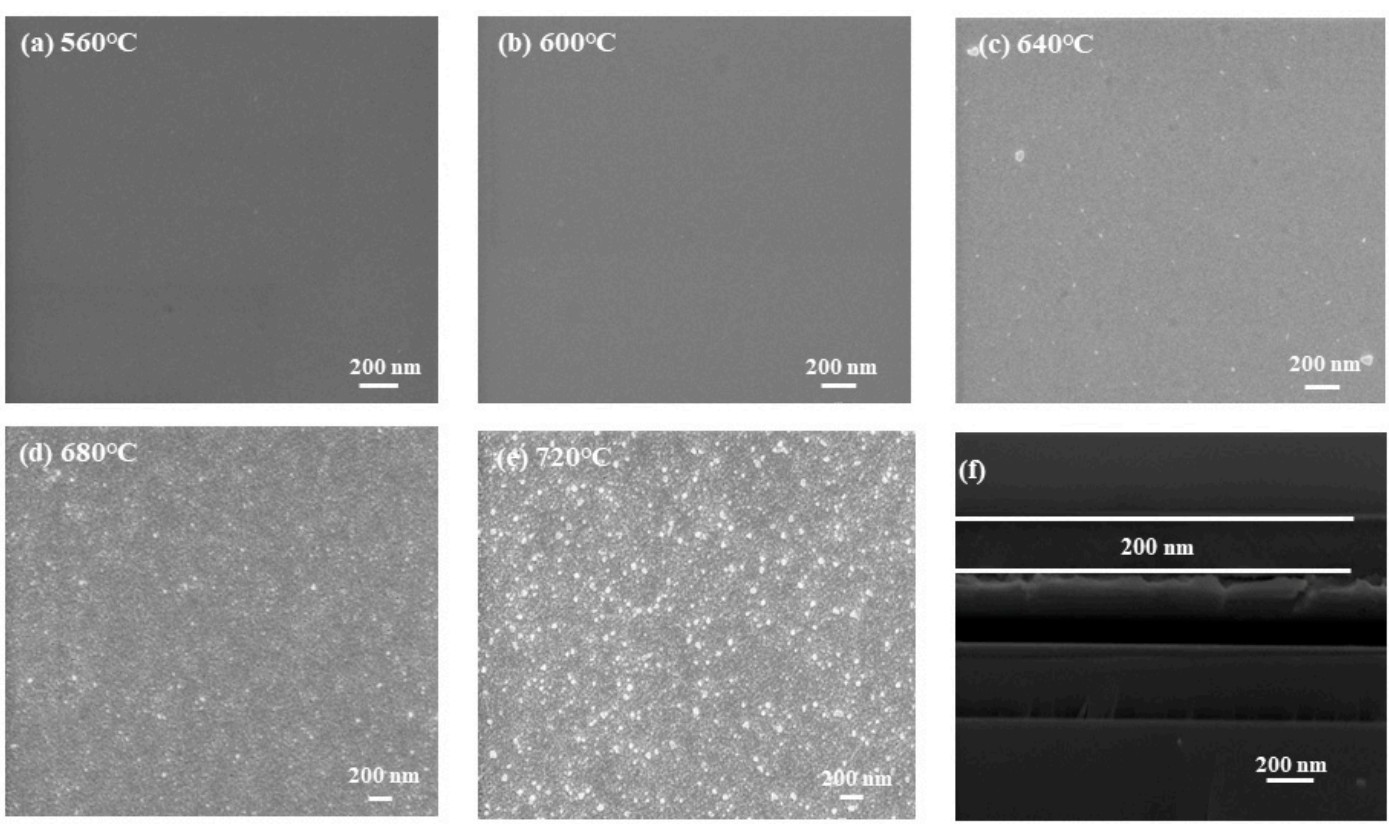

**Figure 2.** SEM micrographs of BT-8%Mn at different annealing temperatures: (**a**) 560 °C; (**b**) 600 °C; (**c**) 640 °C; (**d**) 680 °C; (**e**) 720 °C; (**f**) 640 °C annealed section scan.

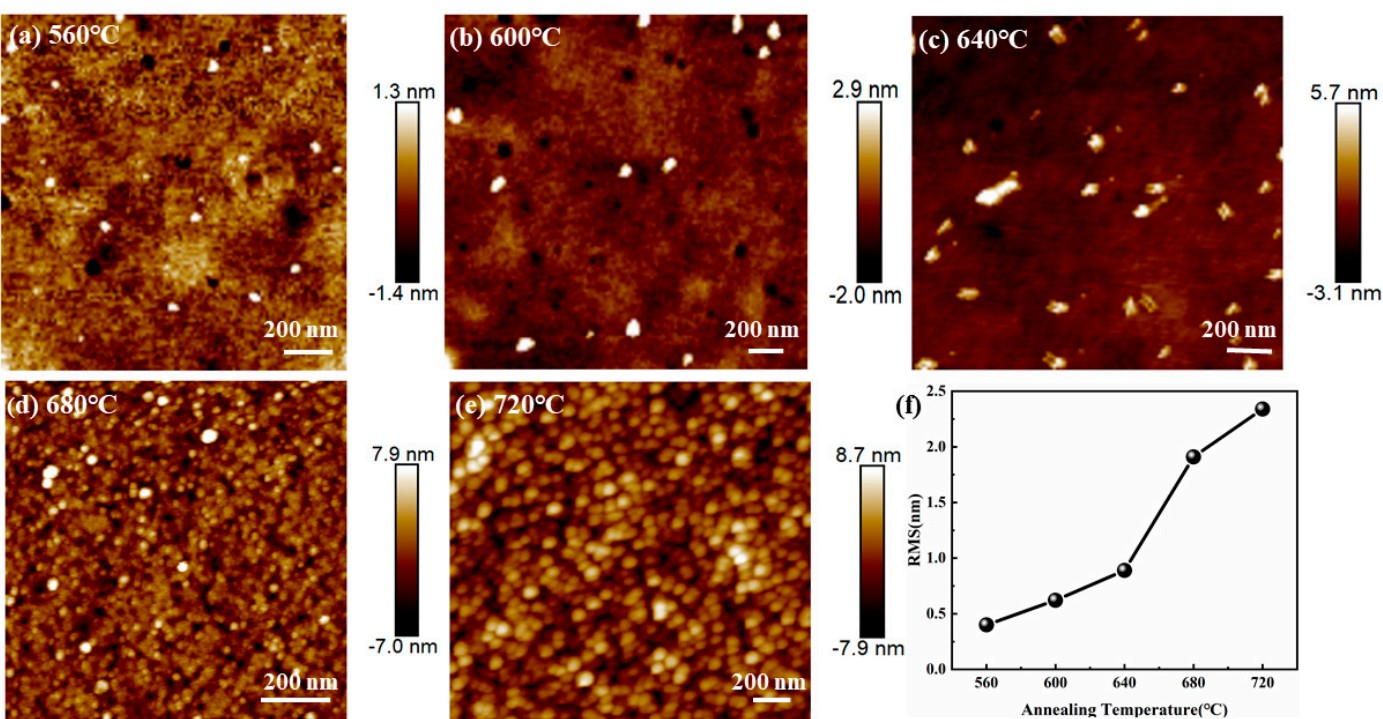

**Figure 3.** AFM images of BT-8%Mn films annealed at different temperatures: (**a**) 560 °C; (**b**) 600 °C; (**c**) 640 °C; (**d**) 680 °C; (**e**) 720 °C; (**f**) variation of RMS with annealing temperature.

Figure 4 shows the EDS (Energy Dispersive Spectrometer) spectrum of BT-8% Mn film, where Figure 4a is the total spectrum of all elements and the rest is the single element

spectrum of the Ba, Ti, O, and Mn elements. It can be seen from the figure that the distribution of each element covers the entire film evenly and that there is no agglomeration. This also reflects the good film preparation process, and the sol-gel method makes it easy to prepare a film with uniform distribution and good quality.

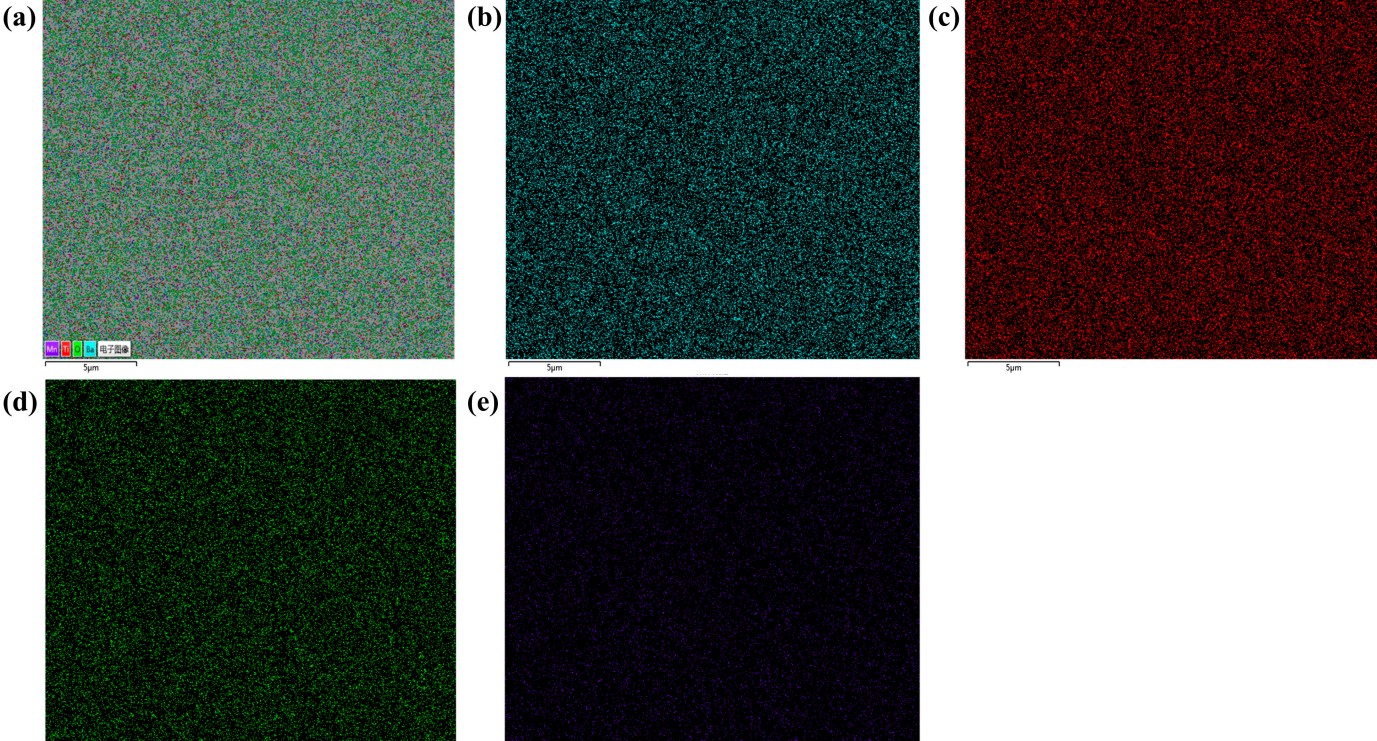

**Figure 4.** BT-8% Mn thin film annealed at 640 °C EDS spectrum: (**a**) element distribution map; (**b**) Ba; (**c**) Ti; (**d**) O; (**e**) Mn.

Figure 5 shows the change in dielectric constant and dielectric loss for films at $10^3$ Hz–$10^6$ Hz with different annealing temperatures. Usually, a high dielectric constant means high crystallinity [38]. The low dielectric constant at low frequencies can be attributed to the fact that the electric field changes slowly at low frequencies, and the response of the dipole polarization mechanism that affects the dielectric constant can keep up with the change in the electric field frequency, so the larger dielectric constant is highlighted. On the contrary, at higher frequencies, most dipoles cannot keep up with the change of the electric field, and the dipole orientation polarization is too late to respond, and the contribution to the dielectric constant is reduced, so the dielectric constant will become smaller [39]. At low annealing temperatures, the dielectric constant of the film is less than 50 due to the majority amorphous phase in the film. When the annealing temperature reaches 680 °C, the dielectric constant is much higher than 50, indicating almost crystalline. The trend of dielectric loss is the same as that of the dielectric constant. Both show a tendency to increase with annealing temperature, which is consistent with what is seen in the microstructure characterization.

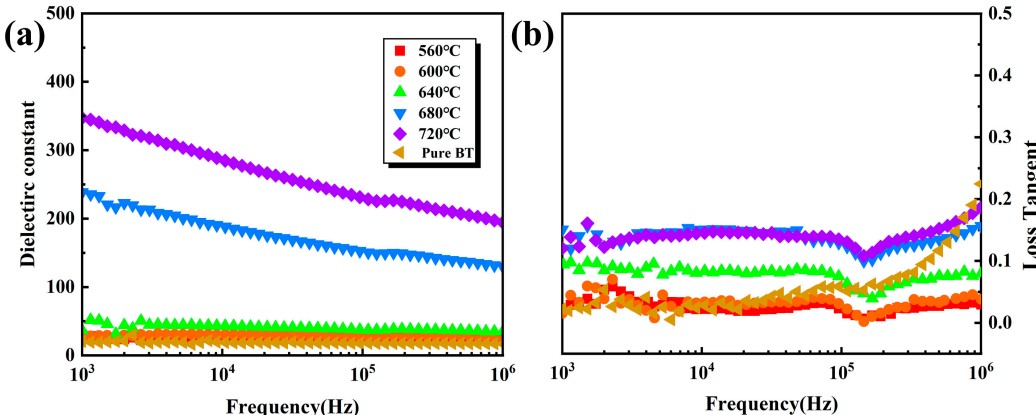

**Figure 5.** Image of BT-8% Mn films dielectric properties with frequency: (**a**) dielectric constant; (**b**) dielectric loss.

Figure 6 shows the temperature dependence of the dielectric constant and the dielectric loss of BT-8% Mn thin films at different annealing temperatures. It can be seen from Figure 6a that the dielectric constant of the film shows a slight increase with temperature at low annealing temperatures (560 °C, 600 °C, 640 °C). This may be due to the fact that as the test temperature increases, the dipole is activated by heat to respond more to the polarization process and increase the dielectric constant. At high annealing temperatures (680 °C, 720 °C), the dielectric constant first increases and then decreases with increasing temperature. The change trend of dielectric loss is basically consistent with the dielectric constant, as shown in Figure 6b. If the dielectric constant of BT-8% Mn film at 20 °C is taken as the reference, the temperature stability of the dielectric constant of the film can be expressed by the formula $\Delta\varepsilon = | \varepsilon - \varepsilon_{20°C}/\varepsilon_{20°C} | \times 100\%$. The results show that the $\Delta\varepsilon$ of all films is less than 15% from 20 °C to 200 °C, which indicates that BT-8% Mn has good dielectric temperature stability.

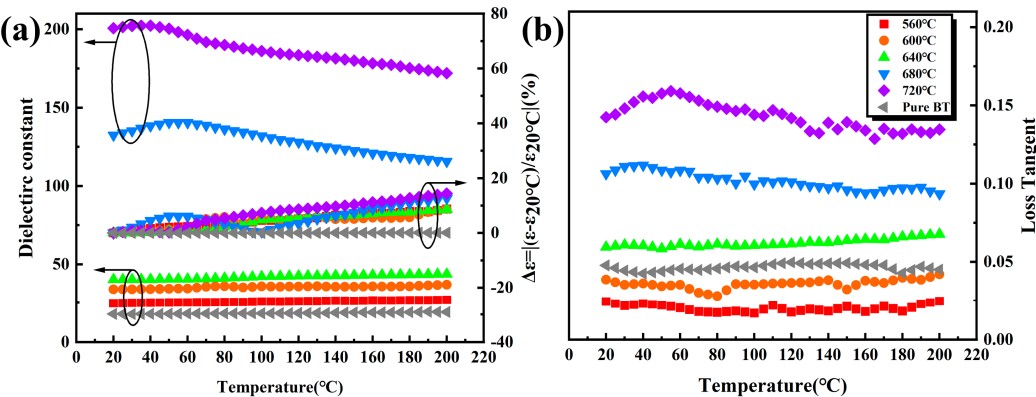

**Figure 6.** Image of BT-8% Mn films dielectric properties with temperature at $10^5$ Hz: (**a**) dielectric constant; (**b**) dielectric loss.

The *P-E* loop is an important parameter reflecting the energy storage performance of the films. Figure 7a shows the *P-E* loop plots of the films at different annealing temperatures. At low temperatures (560 °C, 600 °C, and 640 °C), the P-E loops are thin, indicating excellent energy storage efficiency. As the annealing temperature increases, the P-E loop becomes fatter and $P_r$ increases, then the energy storage efficiency deteriorates sharply. The formula for energy storage density $W = \int_{P_r}^{P_{max}} EdP$ [40] shows that the values of maximum polarization ($P_{max}$) and electric field strength ($E$) should be as large as possible in order to obtain a high energy storage density. Figure 7b shows the $P_{max}$ and its corresponding breakdown strength at different annealing temperatures, limited by the test instrument.

The films prepared at 560 °C, 600 °C, and 640 °C did not break down, so the maximum electric field strength that can be measured was selected as the breakdown strength. At 640 °C, the film obtains a high breakdown strength and a moderate $P_{max}$, which collectively reach the best value for the energy storage density. Figure 7c shows the I–V characteristic curves of the BT-8%Mn films with different annealing temperatures at 0–495 kV/cm electric field strength. It can be observed that the leakage current of the film at high annealing temperatures (680 °C–720 °C) is much larger than at low temperatures (560 °C–640 °C), and the current increases sharply with the increase in the electric field. The high leakage current value not only leads to the low breakdown strength of the film [41], but also causes a sharp decrease in its energy storage efficiency [42]. The trend of energy storage density and energy storage efficiency with annealing temperature is shown in Figure 7d. The highest energy storage density up to 72.4 J/cm³ is for the BT-8%Mn film annealed at 640 °C with 88.5% energy efficiency. The energy efficiency shows a decreasing trend with the increase in annealing temperature, especially after 640 °C, which has a close relationship with crystal phase increase.

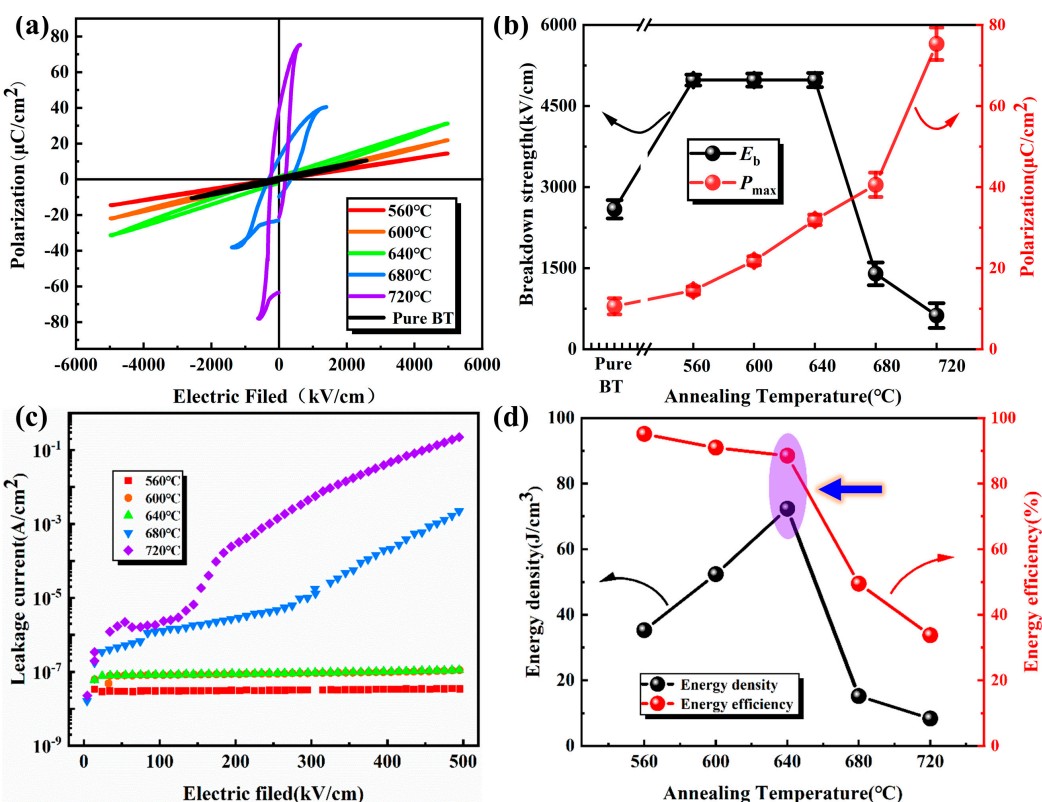

**Figure 7.** BT-8%Mn films at different annealing temperatures: (**a**) *P-E* loop; (**b**) $P_{max}$ and breakdown strength; (**c**) leakage current; (**d**) energy storage density and energy storage efficiency.

Temperature stability is an important parameter to measure the ability of film capacitors to work properly when the environment changes. Figure 8 shows the variation of the film energy storage performance with temperature (20 °C–200 °C) under an electric field of 2489.96 kV/cm. It can be observed that the polarization value increases slightly, and the energy storage efficiency decreases as the temperature increases. The energy storage density fluctuates between 15.05 J/cm³ and 16.58 J/cm³, and the change rate is about 10% as the test temperature of the film increases from 20 °C to 200 °C. Further, the energy storage efficiency decreases from 92.28% to 69.23%. The cycle stability of the film characterizes the change in performance of the film during the long-term charge and discharge process. Through this performance, it can be judged whether the film has a good working life. The cyclic stability of the film is shown in Figure 8c,d. During the charging/discharging cycles

of $10^0$–$10^5$, the polarization value, *P-E* loops' shape, and energy storage performance of the films did not change significantly. This indicates that the film has excellent cycle stability. Excellent temperature and cycle stability enable the film to adapt to harsh conditions and changing environments, and it has a better application prospect.

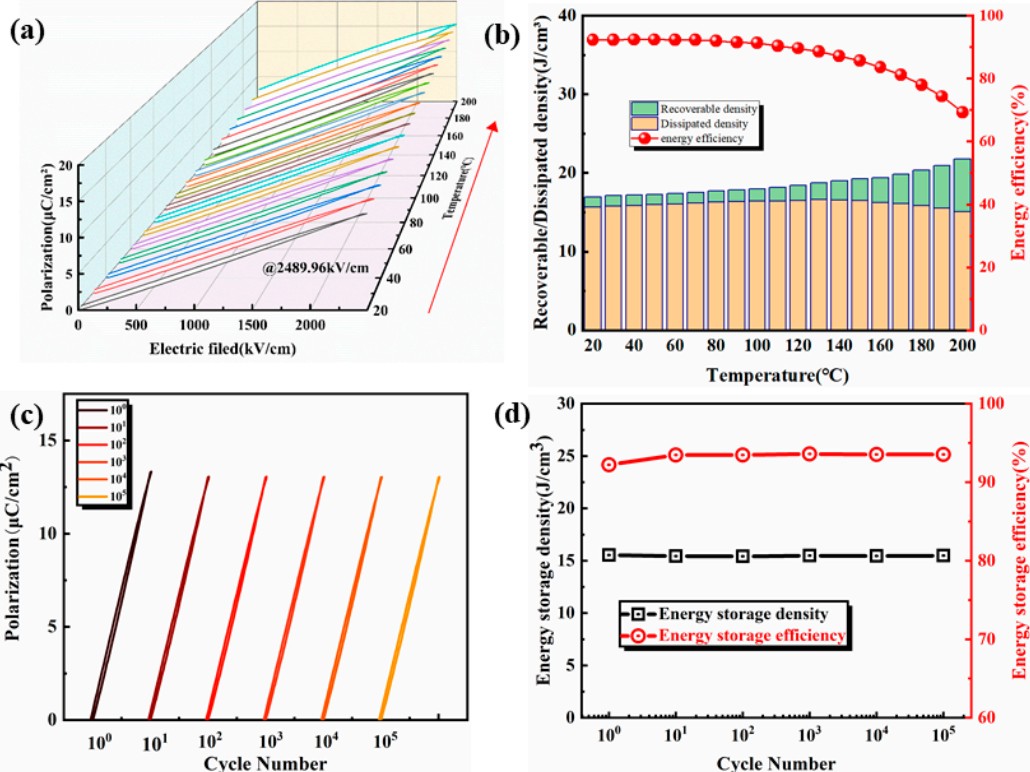

**Figure 8.** (**a**) At 2489.96 kV/cm, *P-E* loops of BT-8%Mn film change with the working temperature; (**b**) Energy storage performance with temperature; (**c**) *P-E* loops change with cycle number of $10^0$–$10^5$; (**d**) Energy storage performance changes with cycle number of $10^0$–$10^5$.

The comparison of the energy-storage performances of BT-8%Mn thin films with other thin films is shown in Figure 9 [43–51]. It can be seen that BT-8%Mn film has excellent energy storage performance, making it a strong competitor with potential in future integrated circuits for energy storage applications.

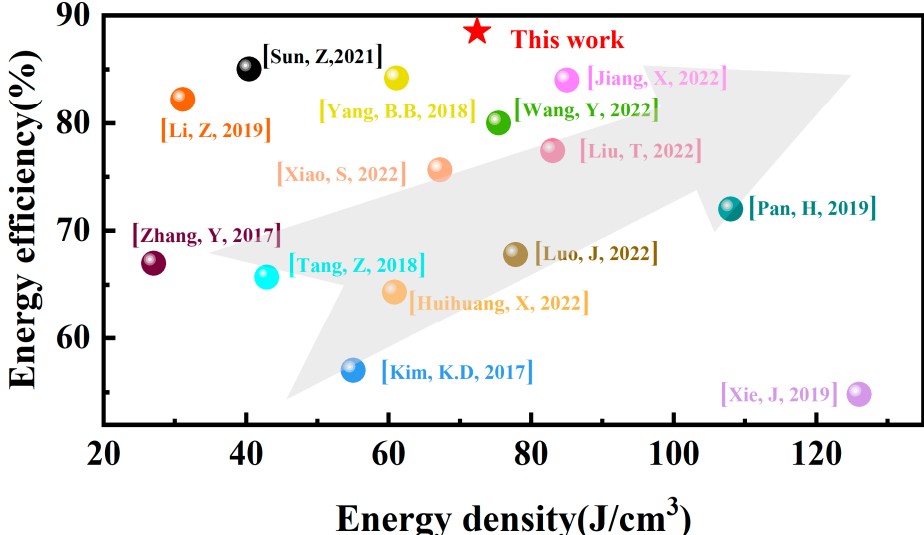

**Figure 9.** Energy storage density and energy storage efficiency of this work compared with other thin films.

## 4. Conclusions

BT films were prepared on $Pt/Ti/SiO_2/Si$ substrates via the sol-gel method with a small amount of Mn-doping and changing the annealing temperature to adjust the crystalline/amorphous phase structures. Multiple methods, including XRD, SEM, AFM, and XPS, demonstrate that Mn could lower the crystallization temperature of BT films, facilitating the formation of an amorphous phase. The coexistence of crystalline/amorphous phases was successfully achieved and characterized by changing the annealing temperature and the excellent energy storage performances: energy density of 72.4 $J/cm^3$ and efficiency of 88.5% are achieved at an annealing temperature of 640 °C. After $10^5$ charge-discharge cycles at 2489.96 kV/cm electric field, the energy storage density and efficiency of the BT-8% Mn film annealed at 640 °C remain basically unchanged, and the change in energy storage density does not exceed 10% in the temperature range of 20 °C–200 °C. It has good cycle stability and temperature stability, which allows it to adapt to multiple charge-discharge cycles and changes in the working environment's temperature. This work proves the effectiveness of constructing a crystalline/amorphous phase by chemical doping and changing the annealing temperature to improve the energy storage performances, which could also be applied in other dielectric systems.

**Author Contributions:** Material fabrication and property characterization, J.G. and D.L.; Writing—original draft, J.G.; Writing—review and editing J.G., D.L., H.H., Q.G., H.X., M.C., Z.Y. and H.L. All authors have read and agreed to the published version of the manuscript.

**Funding:** This research was funded by Guangdong Basic and Applied Basic Research Foundation (No.2022B1515120041, No.2022A1515010073) and Major Program of the Natural Science Foundation of China (Grant No. 51790490).

**Conflicts of Interest:** The authors declare no conflict of interest.

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
