# Peer review of "Tunable Phase Structure in Mn-Doped Lead-Free BaTiO3 Crystalline/Amorphous Energy Storage Thin Films"

_crystals, doi:10.3390/cryst13040649_

Round 1
Reviewer 1 Report
Reviewer
In this manuscript, the authors reported the preparation of lead-free Mn-doped BaTiO3 films on Pt/Ti/SiO2/Si substrates via the sol-gel method and the effects of crystalline/amorphous phase ratio on polarization and electric properties. In general, it works, containing some interesting results. However, multiple aspects related to the description of the analyses and material characterization should be improved before the manuscript is reconsidered for publication.
1. In the section on Material and methods. Why do I use 8% Mn? In stoichiometry, is doping associated with the position of Ba or Ti?
2. XPS experimental details still need to be completed. See the example of the XPS protocol in section 6.1 of Progress in Materials Science 107 (2020) 100591. What was the electron emission angle and the size of the analyzed area? Where samples sputter-etched prior to analyses? If so, what was the Ar+ energy and incidence angle? What was the base pressure during analyses? Was charge neutralizer used? All of these aspects are crucial for the correct interpretation of experimental results.
3. Figure 1c-d - panels still need to be completed, so it is impossible to have a complete interpretation of the surface analysis of the thin film. It is necessary to report the Ti spectrum, which is essential in the electrical response. The clarification will help someone not skilled in XPS.
4. The authors say, “The additional shoulder peak around 531 eV could be considered as an oxygen vacancy after excluding other possibilities (such as carbonate or Al2O3 et al.) [32].” Maybe two or three papers should be added to clarify the peak around 531 eV (for example, https://doi.org/10.1016/j.apsusc.2019.07.003 ---https://doi.org/10.1039/D0CP01010C---https://doi.org/10.1016/j.vacuum.2021.110562).
5. The microstructural details (grain size, porosity level, roughness, etc.) are hardly appreciated in Fig. (2-3). Is the grain size uniform through the thickness of the thin film? Reporting a histogram of the average grain size is needed. What is the relative density of the different grain sizes? These aspects have a strong impact on properties.
6) The strain is an essential parameter with increasing temperature. How does it justify the strain differentiation for thin films with different temperatures?
7) Describe the step by step to determine the energy efficiency.
8) Below are some articles from different authors who contributed to the manuscript.
https://doi.org/10.1016/j.vacuum.2021.110562
https://doi.org/10.1016/S0042-207X(99)00127-X
https://doi.org/10.1111/j.1551-2916.2004.00371.x
https://doi.org/10.1063/1.2357880
https://doi.org/10.1016/j.surfcoat.2006.08.117
Reviewer 2 Report
In the current manuscript “Tunable Phase structure in Mn-doped lead-free BaTiO3 Crystalline/Amorphous energy storage thin films” the authors prepared undoped and Mn doped BaTiO3 film at different temperatures. Obtained pure and doped films in amorphous and crystalline forms were studied.
However, significant part of the measurements of pure BT films (SEM, dielectric measurements, etc) is missing.
Thus, the effect of annealing temperature on Mn-doped BT films without strong comparison with pure BT is described only.
There are following comments for the current manuscript:
1. Describe “SOFCs” in Line 30
2. Explain more why “high polarization characteristics” is important
3. Add values of “high dielectric constant at room temperature” (Line 38), “high dielectric losses” as well as “low withstand voltage”(Line 39) for ceramics and for BT films.
4. Explain why here is film but not ceramic.
5. Add the values of the annealing temperature used for the preparation of the samples mentioned in Lines 43-59.
6. Be sure that the temperature of crystallization for pure BaTiO3 is also included in Introduction.
7. Add information: at what conditions were prepared BT with “hexagonal phase” in mentioned Refs. Add more info about possible phase, at what temperatures, etc.
8. Suppliers of chemicals must be added.
9. What are lattice parameters of pure BT and Mn-BT according to XRD at different temperature?
10. Add information how “breakdown strength” was measured/calculated?
11. What is the thickness of films? 200 nm (Line 132) for all samples (amorphous and crystalline)?
12. How “energy storage” and “energy storage density” were measured/calculated? More details with Eqs. please.
13. Line 132 “The Figure 2 shows The SEM images of BT-8%Mn films annealed at different temperatures.” There are no similar images of pure BT: How it can be compared. 200nm- scale bare must be decreased to see “fine grains”. In this case: Where are “fine grain” size values according to SEM and AFM for pure and doped BT?
14. The same for Fig 3 – can not compare doped BT without pure BT
15. What is the sample in Line 158 and in capture “Figure 4. Image of dielectric properties with frequency:(a) dielectric constant; (b) dielectric loss”. Is Mn-doped BT in Figure 4? Where are dielectric measurements of pure BT? How we can compare? Is it decreased? or increased?
16. The same in Line 168: pure or doped BT?
17. Where is graph “breakdown strength” vs annealing temperature for pure BT? Why reader must believe that Mn-doped BT has better breakdown strength than that of pure BT?
18. What numbers of points/samples were measured to obtain breakdown strength and leakage current values? It must include error bars.
19. What film/process parameters can make influence on breakdown strength (as well as on energy density) and on leakage current (such as film thickness, size of electrode, etc) must be added to the text (may be in Table for comparison similar to Fig. 7).
20. Discussion / comparison must be enlarged.
21. Why 8% of Mn was used (not 5 ot 10)? Add explanation. Probably, TEM image to support the absence of second phase?
22. Add more Refs about pure BT in text (in new Table also) and in Fig 7 for comparison
Round 2
Reviewer 2 Report
The proposed manuscript was edited but still have some questions:
1. Information about the preparation of pure BaTiO3 film (temperature, time, etc) must be added in the text.
2. P-E loop of pure BaTiO3 must be included in Fig.7a for comparison as well as dielectric permittivity and losses of pure BT must be added in Figures 5 and 6, respectively.
3. It is written in coverletter that “Response to comment 10: The breakdown strength is usually calculated by Weibull distribution. However, in this paper, due to the high breakdown strength of the microcrystalline film, the breakdown behavior does not occur in the film annealed at 560 ℃,600 ℃ and 640 ℃ within the measurement range of the instrument. Therefore, the maximum electric field strength is selected as the breakdown strength, which is not expressed by Weibull distribution” and “Response to comment 18: Due to the high breakdown strength of the microcrystalline film, the breakdown behavior does not occur in the film annealed at 560 ℃,600 ℃ and 640 ℃ within the measurement range of the instrument”.
Thus, these three equal values of breakdown strength for films annealed at 560 ℃,600 ℃ and 640 ℃ can be as “approximation” or “speculation” and cannot be used for the further calculation or mentioned as real data. It must be mentioned in the text.
4. Remove “fine” grains.
5. Add pure BT to “breakdown strength” graph in Fig 7b.
6. Only three tests cannot guarantee correct data. Moreover, authors present here “Response to comment 18: … Therefore, the maximum electric field strength is selected as the breakdown strength.” It is not correct. The average values after at least 10 measurements with the error bars must be presented.
